# Changes in the Kinematics of Midfoot and Rearfoot Joints with the Use of Lateral Wedge Insoles

**DOI:** 10.3390/jcm11154536

**Published:** 2022-08-03

**Authors:** Álvaro Gómez Carrión, Maria de los Ángeles Atín Arratibe, Maria Rosario Morales Lozano, Carmen Martínez Rincón, Carlos Martínez Sebastián, Álvaro Saura Sempere, Almudena Nuñez-Fernandez, Rubén Sánchez-Gómez

**Affiliations:** Nursing Department, Faculty of Nursing, Physiotherapy and Podiatry, Universidad Complutense de Madrid, 28040 Madrid, Spain; alvaroalcore@hotmail.com (Á.G.C.); matin@ucm.es (M.d.l.Á.A.A.); rmorales@ucm.es (M.R.M.L.); nutrias@ucm.es (C.M.R.); carlos_mar_seb@hotmail.com (C.M.S.); alvarosaura@gmail.com (Á.S.S.); almnun01@ucm.es (A.N.-F.)

**Keywords:** midfoot joint, rearfoot joint, navicular, calcaneus, lateral wedge insoles, Polhemus device, tracking device

## Abstract

The lateral wedge insole (LWI) is a typical orthopedic treatment for medial knee osteoarthritis pain, chronic ankle instability, and peroneal tendon disorders. It is still unknown what the effects are in the most important joints of the foot when using LWIs as a treatment for knee and ankle pathologies. **Objectives:** The aim of this study was to determine the influence of LWIs on the position of the midfoot and rearfoot joints by measuring the changes using a tracking device. **Methods:** The study was carried out with a total of 69 subjects. Movement measurements for the midfoot were made on the navicular bone, and for the rearfoot on the calcaneus bone. The Polhemus system was used, with two motion sensors fixed to each bone. Subjects were compared by having them use LWIs versus being barefoot. **Results:** There were statistically significant differences in the varus movement when wearing a 4 mm LWI (1.23 ± 2.08°, *p* < 0.001) versus the barefoot condition (0.35 ± 0.95°), and in the plantarflexion movement when wearing a 4 mm LWI (3.02 ± 4.58°, *p* < 0.001) versus the barefoot condition (0.68 ± 1.34°), in the midfoot. There were also statistically significant differences in the valgus movement when wearing a 7 mm LWI (1.74 ± 2.61°, *p* < 0.001) versus the barefoot condition (0.40 ± 0.90°), and in the plantar flexion movement when wearing a 4 mm LWI (2.88 ± 4.31°, *p* < 0.001) versus the barefoot condition (0.35 ± 0.90°), in the rearfoot. **Conclusions:** In the navicular bone, a varus, an abduction, and plantar flexion movements were generated. In the calcaneus, a valgus, an adduction, and plantar flexion movements were generated with the use of LWIs.

## 1. Introduction

Lateral wedge insoles (LWIs) are specific orthopedic treatments indicated to treat several lateral foot pathologies, such as ankle sprains [1]. An ankle sprain is a common injury in competitive athletes in both sports and military activities [2]. Ankle sprains account for 7.3% of all college sports injuries [3], and for the highest share of ultra-endurance racing injuries (at 28.6%) [4]. The peroneal tendons are responsible for stabilizing the ankle and preventing the inversion of the foot [5]; therefore, they can also be injured during an ankle sprain [6]. When performing sports activities where there are jumps, sprints, or landings, the ankle and rearfoot are placed in plantar flexion and varus positions. These two positions can cause an ankle sprain; thus, LWIs could be recommended to avoid them [6,7,8]. It has been shown that a “high vertical peak of ground reactive forces” is considered to be a factor in the appearance of sprains [9,10]. Furthermore, when there is a greater height of the midfoot, this vertical peak GRF is greater than that for the lower midfoot [11]. The navicular bone is the most important part of the arch of the foot, forming part of the midfoot—specifically by forming the talonavicular joint [12]. Navicular height is used as a risk factor in multiple pathologies, such as medial tibial stress [13], patellofemoral pain [14], posterior tibial dysfunction [15], plantar fasciitis [16], and ankle sprain [17]. Additional considered factors in lateral sprains include an oversupinated foot and a calcaneus varus position [18]. LWIs have also shown changes in electromyographic activity in the peroneal tendons [1,19]. They have also been shown to generate a valgus movement in the subtalar joint by reducing calcaneus and knee varus [20]. This orthopedic treatment has demonstrated its efficacy in reducing medial osteoarthritis pain. An LWI can laterally displace the center of mass and decrease knee adduction. This change produces a decrease in pressure on the medial compartment of the knee and, therefore, improves function [21].

The effectiveness of different thicknesses of LWIs in reducing pain and varus moments of the knee has been proven. However, their effects on the foot are still unknown—specifically, the effects on the midfoot and rearfoot joints. The movement of these parts of the foot was evaluated with the 6 SpaceFastrak system tracking device, used in previous studies to measure the changes in position of the navicular and calcaneus bones [22,23,24,25]. This device can evaluate the movement of the navicular and calcaneus bones with precision [26]. Blackwood used this device to test the classical theory of the functioning of the midfoot. He observed how the position of the calcaneus bone affected the movement of the forefoot, resulting in greater movement of the forefoot with the rearfoot in the valgus position [22].

Due to the interest in knowing the biomechanical effects of LWIs on the foot, the scientific goal of the present study was to determine the effects of lateral wedge insoles on the midfoot and rearfoot of the subjects.

## 2. Materials and Methods

### 2.1. Participants

The subjects were informed of the objectives of the study and were asked to read and sign their informed consent to participate. The ethical requirements of the Declaration of Helsinki were followed. The present study obtained the approval of the ethics committee of the Nta Sra de Valme University Hospital (Seville), with the registration code 1541-N-20.

The study design consisted of comparing the use of lateral wedge insoles vs. being barefoot by recording the movement of the navicular and calcaneus bones. The Polhemus Fastrak Patriot [25] was used for the measurement of the navicular and calcaneus bones, which resulted in a significant intraclass correlation coefficient of 0.996 (95% CI) with the use of a 7 mm rearfoot varus wedge in this study.

The statistical unit of the public University Complutense of Madrid calculated the sample for the present study. The results of the study in the navicular bone (0.35 ± 0.95° to 1.35° ± 2.41°) (*p* < 0.001) were used for the calculation of the sample [25]. Taking into consideration the need for a confidence interval (CI) of 95%, with a statistical power of 80%, _x = 0.05, and _b = 20%, and assuming the common loss of 20% of the participants, 86 subjects needed to be included in this study, and 69 subjects were ultimately enrolled.

All subjects needed to meet the following inclusion criteria: (1) subjects aged between 18 and 65 years, and of both sexes; (2) neutral foot results in the foot posture index [27]; (3) subjects had to be in a standing position and have a foot length between 25 cm and 27 cm. There were two exclusion criteria to this study: The first was if the subject had undergone a prior surgery on the foot and/or the lower limb. The second exclusion criterion was if the subject had congenital deformities of the lower limbs (e.g., foot deformity, varus valgus knees, discrepancy in the length of the limb) [28]. The subjects were obtained from the Podos Clinic (Seville) from October 2020 to January 2022.

### 2.2. Procedures

The study of the movement of the navicular and the calcaneus bones was carried out using the 6 SpaceFastrak system tracking device (Polhemus Inc., Colchester, VT, USA). This system consists of three parts: a transmitter, and two sensors. The 120 Hz transmitter (Figure 1) locates and records the signal from the sensors. Then, it is digitally transformed by the FT3HostSWCD-2.1.0 software, and finally gives a result of the spatial orientation variables of each sensor (Table 1). Two sensors were used: sensor one was placed to align with the back of the calcaneus bone, and sensor two was placed to align with the medial navicular tubercle. The calcaneus bone sensor was used as a proximal motion reference, and the navicular bone sensor was used as a distal motion reference (Figure 2).

The protocol began by measuring the foot posture index barefoot in a standing position. If a result from 0 to +5 was obtained, the subject was included in the study. Then, the area of the navicular and the calcaneus bones where the sensor was placed was distinguished with a red marker.

To prevent the movement of the sensor, a Hypafix bandage was used. The software incorporated into the Polhemus Fastrak system was used for data processing. The subject was placed in a relaxed standing position, and the calibration of each sensor was recorded, taking an initial reference. The subject was asked not to make movements and not to carry metal elements to avoid distorting the data. This protocol was based on Corwall’s works [26,29]. First, measurements of the calcaneus and navicular bones were taken with the subjects being barefoot. These measurements were repeated three times. Then, the next measurements of the navicular and calcaneus bones were taken with the LWIs. Three different thicknesses of LWIs were used: 4 mm, 7 mm, and 10 mm. These measurements were repeated a total of three times each to avoid errors in the results. A wedge of the same thickness was always used on the contralateral foot to avoid imbalances (Figure 3).

### 2.3. Lateral Wedge Insoles

The insoles were made by the principal investigator (A.aa), who has 10 years of experience. The same pink top cover was used in all LWIs of all thicknesses to avoid identifying the amount of correction it had. The wedge was composed of a Shore 70A material of ethylene-vinyl acetate (EVA). The wedges had thicknesses of 4 mm, 7 mm, and 10 mm, based on previous studies [30,31,32,33,34,35,36,37].

### 2.4. Statistical Analysis

The statistical unit of the public University Complutense of Madrid performed the statistical analysis using SPSS Version 20.0 for Windows (IBM Corp., Armonk, NY, USA) in order to evaluate whether the data were normally distributed. The Kolmogorov–Smirnov test was used, with a result of a non-normal distribution (*p* < 0.05). A nonparametric paired Friedman test was used to correct for multiple comparisons of *p*-values, and to show that the LWIs’ variables were different. The Wilcoxon test with bivariate correlations was used to determine whether significant movement changes were detected between the four conditions.

## 3. Results

For the study, we initially selected a total of 86 subjects; of these, 17 subjects did not meet the inclusion criteria. In total, 69 subjects (36 men and 33 women) were recruited for this study. The measurements of barefoot subjects were used as the control group. Table 2 shows the sociodemographic characteristics of the participants, and a normal distribution was obtained for each (*p* > 0.05)

The reliability of the data is shown with the intraclass correlation coefficients (ICCs), standard measurement error (SEM), and minimum detectable change (MDC) of the two sensors under the different conditions in Table 3 and Table 4. For the calcaneus bone, the ICC has an interval of 0.998 to 0.862. For the navicular bone, it has an interval of 0.995 to 0.831. According to the Landis and Koch classification criteria of the ICCs, a result greater than 0.81 indicates perfect reliability [38]. The MDC for the calcaneus bone has an interval of 1.644° to 0.144°, while for the navicular bone it has an interval of 1.888° to 0.115°. The SEM for the calcaneus bone has an interval of 0.593° to 0.056°, while for the navicular bone it has an interval of 0.681° to 0.066°.

The data of the navicular bone sensor movement under the four conditions are shown in Table 5 and Figure 4.

Wearing an LWI caused an abduction displacement in the navicular bone. This was only statistically significant with LWI10, with an average range of NAVIC-ABDUC motion of 2.01 ± 2.95° (95% CI of 1.28–2.78) (*p* < 0.05). The displacement of the plantar flexion was statistically significant with the use of LWIs. With LWI4, the average displacement recorded in NAVIC-PLANFLEX was 3.02 ± 4.58° (95% CI of 2.04–4.36) (*p* < 0.001). With LWI7, the average displacement recorded in NAVIC-PLANFLEX was 2.51 ± 3.55° (95% CI of 1.60–3.42) (*p* < 0.05). With LWI10, the average displacement recorded in NAVIC-PLANFLEX was 2.48 ± 3.69° (95% CI of 1.54–3.42) (*p* < 0.001).

The next remarkable results concerning the navicular bone were the varus and valgus displacements caused by LWIs. The average displacement recorded from the navicular bone in NAVIC-VAR was 1.23 ± 2.08° (95% CI of 0.87–1.70) (*p* < 0.001), and the average displacement recorded in NAVIC-VAL was 1.04 ± 1.98° (95% CI of 0.53–1.54) (*p* < 0.05). With LWI7, the average displacement recorded from the navicular bone in NAVIC-VAR was 1.23 ± 1.60° (95% CI of 0.84–1.58) (*p* < 0.001), and the average displacement recorded in NAVIC-VAL was 0.96 ± 1.71° (95% CI of 0.53–1.40) (*p* < 0.05). With LWI10, the average displacement recorded from the navicular bone in NAVIC-VAR was 0.98 ± 1.41° (95% CI of 0.62–1.50) (*p* < 0.001), and the average displacement recorded in NAVIC-VAL was 1.16 ± 1.73° (95% CI of 0.71–1.60) (*p* < 0.001).

The data of the calcaneus bone sensor movement under the four conditions are shown in Table 6 and Figure 5.

Wearing an LWI caused an adduction displacement in the calcaneus bone. With LWI4, the average displacement recorded in CALCA-ADDUC was 1.76 ± 2.66° (95% CI of 1.08–2.44) (*p* < 0.001). With LWI7, the average displacement recorded in CALCA-ADDUC was 1.83 ± 2.42° (95% CI of 1.21–2.46) (*p* < 0.001). With LWI10, the average displacement recorded in CALCA-ADDUC was 1.91 ± 2.81° (95% CI of 1.19–2.63) (*p* < 0.001).

The displacement of the plantar flexion position was statistically significant when wearing LWIs. With LWI4, the average displacement recorded in CALCA-PLANFLEX was 2.88 ± 4.31° (95% CI of 1.76–3.98) (*p* < 0.001). With LWI7, the average displacement recorded in CALCA-PLANFLEX was 3.07 ± 4.07° (95% CI of 2.01–4.11) (*p* < 0.001). With the LWI10, the average displacement recorded in CALCA-PLANFLEX was 2.61 ± 3.56° (95% CI of 1.70–3.53) (*p* < 0.001).

The next remarkable results for the calcaneus bone were the varus and valgus displacement caused by the LWIs. The average displacement recorded from the calcaneus bone in CALCA-VAR was 1.30 ± 1.70° (95% CI of 0.86–1.77) (*p* < 0.05). With LWI7, the average displacement recorded from the calcaneus bone in CALCA-VAR was 1.21 ± 1.45° (95% CI of 0.54–1.57) (*p* < 0.001), while that in CALCA-VAL was 1.74 ± 2.61° (95% CI of 1.06–2.41) (*p* < 0.001). With LWI10, the average displacement recorded from the calcaneus bone in CALCA-VAR was 1.06 ± 1.69° (95% CI of 0.62–1.49) (*p* < 0.001), while that in CALCA-VAL was 1.47 ± 2.50° (95% CI of 0.83–2.11) (*p* < 0.001). No further data were obtained with relevant changes in significance.

## 4. Discussion

The present study aimed to evaluate the effects of the different thicknesses of LWIs on the rearfoot and midfoot. The results of this study show that the lateral wedge insoles generated abduction displacement in the navicular bone (*p* < 0.05), in addition to the plantar flexion displacement (*p* < 0.001). The results also showed how the LWIs caused the majority of varus displacement in the navicular bone (*p* < 0.001). For the calcaneus bone, the results of this study show that the lateral wedge insoles generated adduction displacement (*p* < 0.001) and plantar flexion displacement (*p* < 0.001). In addition, the LWIs caused the majority of valgus displacement (*p* < 0.001).

The results obtained in this study show that when LWIs were used, Root’s theory was fulfilled. According to this classic theory, when the rearfoot is in a valgus position, the navicular bone is in a varus position [39].

The use of LWIs for pathologies of medial osteoarthritis of the knee has been broadly researched over the last few years. Favorable results have been collected in terms of the reduction in the severity of knee pain with their use in various studies, such as those by Felson [40] and Hunt [41]. A decrease in knee adduction movement was also obtained in studies by Hinman [42] and Butler [43]. LWIs were not only used for knee pathologies, but were also used to treat other pathologies because of their improvement of symptoms in the peroneal muscles. Bahur [1] demonstrated an improvement in the pre-activation of the long peroneal muscle with the use of 3 mm LWIs. The results obtained by Sánchez-Gómez [19] show that the use of 3 mm, 6 mm, and 9 mm LWIs reduced the activation of the long peroneal muscle.

We are not able to compare our results with those of other studies on navicular bone movement while wearing lateral wedge insoles, because there has not been any prior research on these technical characteristics. We only found a cadaveric study by Blackwood [22] in line with our findings relating to plantarflexion of the navicular bone. Blackwood’s findings show that the valgus position of the calcaneus bone increases the movement of the first metatarsal bone in the sagittal plane. This circumstance could be linked with medial longitudinal arch collapse, due to the increase in dorsiflexion of the first metatarsal bone [9,26].

Regarding the movement of the calcaneus bone with the use of lateral wedge insoles, our results of displacement in the valgus resemble those described by other authors, such as Kakihana [20] and Abdallah [44]. The wearing of these wedges generated significant valgus displacement in the subtalar joint in both studies. In Chapman’s study [45], wearing LWIs caused an external movement of the pressure center of the foot, causing a greater eversion of the foot. These results were also observed by Kakihana [20], who used a 6° lateral wedge and obtained a significant increase in the valgus moment of the subtalar joint (*p* < 0.001) while reducing the varus moment of the knee joint (*p* < 0.001). Hatfield’s [46] results show that the adduction moment of the knee was reduced by 8% with the LWI, and by 6% when using an LWI with an arch support (*p* < 0.05). Hatfield’s findings also show that between using an LWI and using an LWI with an arch support, the LWI alone was also associated with a more everted foot position (4.3°) than with the arch support (3.2°) (*p* < 0.05). Nester’s [47] study revealed how anti-supinator orthoses acted by increasing pronation during the contact phase, as well as the total range of the rearfoot complex. Another study by Nester [48] showed that wearing LWIs generated more eversion in the rearfoot, less ground-reactive forces in the contact phase, and better cushioning. In Souza’s study [49], the use of a sandal with a 5° lateral wedge increased the eversion of the rearfoot during the midstance phase (*p* < 0.05). It also showed that using a sandal with a side wedge of 10° increased the eversion of the rearfoot during the midstance and takeoff phases (*p* < 0.05), as compared to the use of flat sandals. Lin et al. [50] obtained an increase in plantar flexion of the ankle with the use of rearfoot varus wedges. In our study, we found the same movement changes in the plantar flexion of the navicular and calcaneus bones with the use of LWIs.

Kakihana [30] and Uto [31] demonstrated the effects of LWIs in terms of laterally shifting the center of pressure on the ankle and the knee, respectively. This caused an increase in the rearfoot valgus, obtaining results in accordance with those described in our research. Increased valgus movement protects the ankle from the pathology of lateral instability, for which the varus position of the rearfoot is a risk factor [18].

According to the present results, the prescription of these lateral wedge insole treatments is safe to treat medial osteoarthritis of the knee and lateral instability of the ankle. The use of LWIs does not alter the arch structures, because the valgus effect is not present on the navicular bone.

### Limitation

This study has some limitations. Unwanted movements were generated during the attachment of the sensors to the skin or when the wedges were changed. The subjects took time to adapt to the wedges. The LWIs possibly generated instability during the data collection, even with respect to the time required to take the measurements. Sometimes, a dispersion of data was found in the sample due to the high sensitivity of the instrument.

## 5. Conclusions

The data obtained in the present study show that the use of lateral wedge insoles can generate changes in the position of the navicular and calcaneus bones during weight-bearing. A varus, an abduction, and plantar flexion movements were generated specifically in the midfoot joint, represented by the navicular bone. A valgus, an adduction, and plantar flexion movements were generated with the use of LWIs in the rearfoot joint, represented by the calcaneus bone. After knowing these effects on the midfoot and the rearfoot, it is safe to prescribe LWIs to treat medial osteoarthritis pathologies and lateral ankle instabilities.

## Figures and Tables

**Figure 1 jcm-11-04536-f001:**
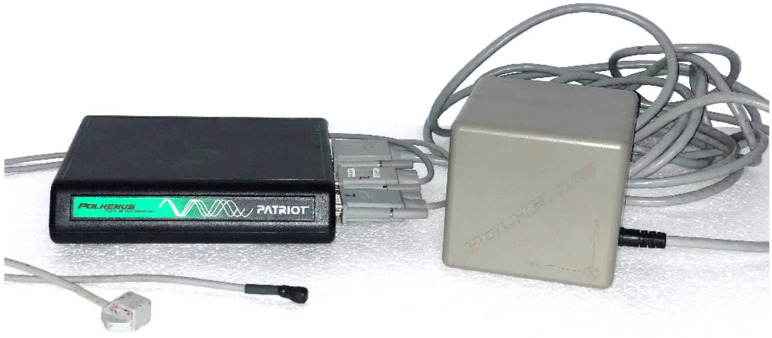
Polhemus Fastrak system. From left to right: receiver module, two sensors, emitter module.

**Figure 2 jcm-11-04536-f002:**
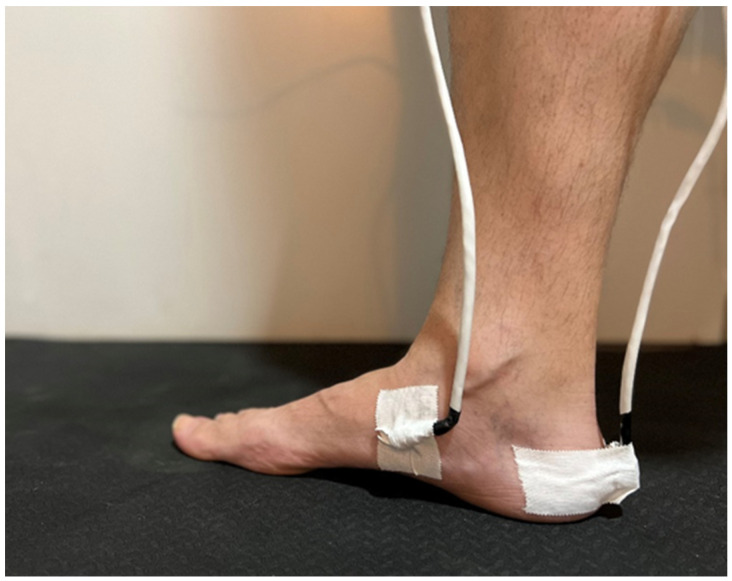
Sensor one was placed on the posterior part of the calcaneus bone, and sensor two was placed on the medial navicular tubercle.

**Figure 3 jcm-11-04536-f003:**
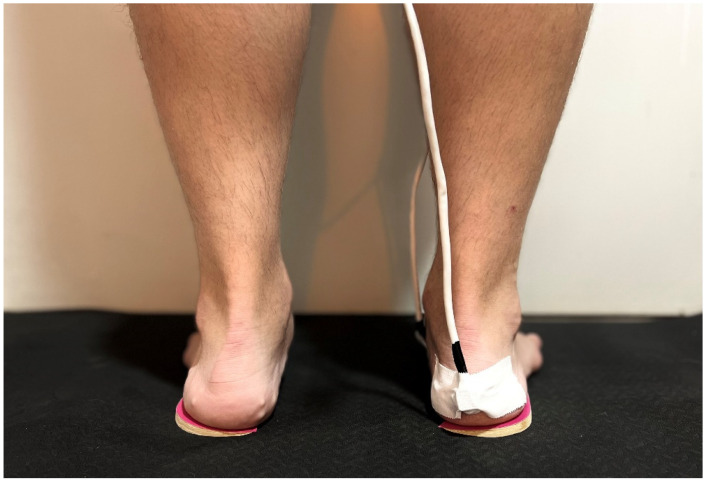
LWIs were placed under the calcaneus bones of both feet.

**Figure 4 jcm-11-04536-f004:**
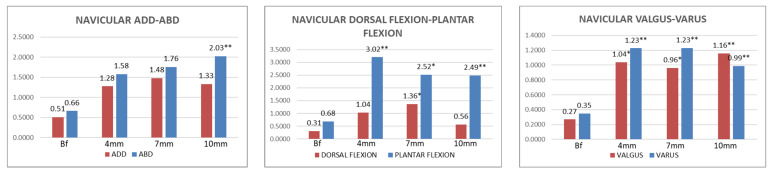
Graphical summary of the degrees of movement in adduction, abduction, dorsal flexion, plantar flexion, valgus, and varus for the navicular bone in four different conditions. Abbreviations: Bf = barefoot; 4 mm = lateral insoles, 4 mm; 7 mm = lateral insoles, 7 mm; 10 mm = lateral insoles, 10 mm. *p*-Value = level of significance; *p* < 0.05 * (with a 95% confidence interval) was considered statistically significant, and *p* < 0.001 ** (with a 95% confidence interval) was considered strongly statistically significant.

**Figure 5 jcm-11-04536-f005:**
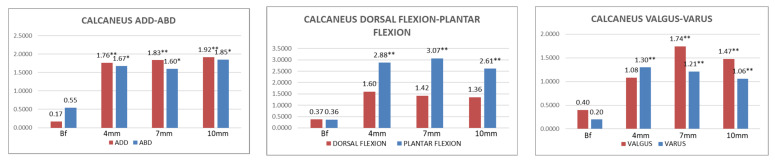
Graphical summary of the degrees of movement in adduction, abduction, dorsal flexion, plantar flexion, valgus, and varus for the calcaneus bone in four different conditions. Abbreviations: Bf = barefoot; 4 mm = lateral insoles, 4 mm; 7 mm = lateral insoles, 7 mm; 10 mm = lateral insoles, 10 mm. *p*-Value = level of significance; *p* < 0.05 * (with a 95% confidence interval) was considered statistically significant, and *p* < 0.001 ** (with a 95% confidence interval) was considered strongly statistically significant.

**Table 1 jcm-11-04536-t001:** The table shows the three-dimensional space variables.

**Navicular**		
Axis (X):	NAVIC-DORFLEX	NAVIC-PLANFLEX
Axis (Y):	NAVIC-VAR	NAVIC-VAL
Axis (Z):	NAVIC-ADDUC	NAVIC-ABDUC
**Calcaneus**		
Axis (X):	CALCA-DORFLEX	CALCA-PLANFLEX
Axis (Y):	CALCA-VAR	CALCA-VAL
Axis (Z):	CALCA-ADDUC	CALCA-ABDUC

Abbreviations: NAVIC-DORFLEX = navicular dorsal flexion; NAVIC-PLANFLEX = navicular plantar flexion; NAVIC-VAR = navicular varus; NAVIC-VAL = navicular valgus; NAVIC-ADDUC = navicular adduction; NAVIC-ABDUC = navicular abduction; CALCA-DORFLEX = calcaneus dorsal flexion; CALCA-PLANFLEX = calcaneus plantar flexion CALCA-VAR = calcaneus varus; CALCA-VAL = calcaneus valgus; CALCA-ADDUC = calcaneus adduction; CALCA-ABDUC = calcaneus abduction.

**Table 2 jcm-11-04536-t002:** Characteristics of subject groups.

Variable	*n* = 60 Mean ± SD (95% CI)
Age (years)	28.41 ± 8.89 (24.79–31.10)
FPI (scores)	1.71 ± 1.47 (1.27–1.96)
Weight (kg)	65.81 ± 12.68 (62.32–71.50)
Height (cm)	166.74 ± 11.84 (163.56–171.09)

Abbreviations: SD = standard deviation; CI = confidence interval (with a 95% confidence interval); FPI = foot posture index.

**Table 3 jcm-11-04536-t003:** The reliability of data for ICC, SEM, and MDC of the navicular bone in barefoot condition and wearing lateral wedge insoles.

			BARE			LWI4				LWI7				LWI10		
Variables	SD	ICC	SEM	MDC	SD	ICC	SEM	MDC	SD	ICC	SEM	MDC	SD	ICC	SEM	MDC
		(95%CI)				(95%CI)				(95%CI)				(95%CI)		
NAVIC-ADDUC	0.779	0.831	0.32	0.887	2.174	0.981	0.303	0.841	2.485	0.993	0.197	0.546	2.272	0.992	0.208	0.577
		(0.740–0.894)				(0.970–0.988)				(0.990–0.996)				(0.987–0.995)		
NAVIC-ABDUC	1.700	0.85	0.659	1.826	2.653	0.937	0.666	1.845	2.818	0.989	0.29	0.8045	2.957	0.984	0.378	1.050
		(0.770–0.905)				(0.903–0.960)				(0.984–0.993)				(0.975–0.990)		
NAVIC-PLANFLEX	1.340	0.841	0.535	1.483	4.581	0.983	0.589	1.632	3.559	0.988	0.385	1.069	3.695	0.981	0.517	1.435
		(0.754–0.900)				(0.975–0.990)				(0.982–0.993)				(0.970–0.988)		
NAVIC-DORFLEX	0.458	0.979	0.066	0.184	1.973	0.985	0.246	0.68	2.431	0.921	0.681	1.886	1.226	0.978	0.179	0.498
		(0.968–0.987)				(0.976–0.990)				(0.880–0.951)				(0.967–0.986)		
NAVIC-VAR	0.947	0.985	0.115	0.32	2.082	0.995	0.152	0.421	1.590	0.986	0.186	1.887	1.417	0.994	0.112	0.311
		(0.977–0.991)				(0.992–0.997)				(0.979–0.991)				(0.990–0.996)		
NAVIC-VAL	0.385	0.831	0.158	0.439	1.975	0.991	0.182	0.505	1.717	0.978	0.251	1.888	1.737	0.992	0.153	0.426
		(0.741–0.893)				(0.987–0.995)				(0.967–0.987)				(0.988–0.995)		

Abbreviations: SD = standard deviation; ICC = intraclass correlation coefficient; SEM = standard error of measurement; MDC = minimal detectable change; CI = confidence interval; NAVIC-ADDUC = navicular adduction; NAVIC-ABDUC = navicular abduction; NAVIC-PLANFLEX = navicular plantar flexion; NAVIC-DORFLEX = navicular dorsal flexion; NAVIC-VAR = navicular varus; NAVIC-VAL = navicular valgus; BARE = barefoot; LWI4 = lateral wedge insoles, 4 mm; LWI7 = lateral wedge insoles, 7 mm; LWI10 = lateral wedge insoles, 10 mm.

**Table 4 jcm-11-04536-t004:** The reliability of data for ICC, SEM, and MDC of the calcaneus bone in barefoot condition and wearing lateral wedge insoles.

			BARE			LWI4				LWI7				LWI10		
Variables	SD	ICC	SEM	MDC	SD	ICC	SEM	MDC	SD	ICC	SEM	MDC	SD	ICC	SEM	MDC
		(95%CI)				(95%CI)				(95%CI)				(95%CI)		
CALCA-ADDUC	0.307	0.967	0.056	0.155	2.662	0.995	0.179	0.497	2.422	0.997	0.131	0.365	2.814	0.995	0.195	0.541
		(0.948–0.979)				(0.993–0.997)				(0.995–0.998)				(0.993–0.997)		
CALCA-ABDUC	1.062	0.862	0.394	1.093	2.793	0.955	0.593	1.644	2.275	0.987	0.259	0.721	3.012	0.997	0.159	0.442
		(0.788–0.913)				(0.931–0.972)				(0.980–0.992)				(0.996–0.998)		
CALCA-PLANFLEX	0.900	0.883	0.307	0.852	4.315	0.998	0.189	0.524	4.068	0.996	0.231	0.639	3.558	0.998	0.146	0.405
		(0.821–0.926)				(0.997–0.999)				(0.995–0.998)				(0.997–0.999)		
CALCA-DORFLEX	0.796	0.891	0.263	0.729	2.772	0.996	0.176	0.488	2.337	0.993	0.186	0.517	2.603	0.996	0.164	0.455
		(0.833–0.931)				(0.994–0.997)				(0.990–0.996)				(0.994–0.997)		
CALCA-VAR	0.484	0.896	0.156	0.433	1.689	0.989	0.181	0.499	1.458	0.979	0.209	0.581	1.686	0.995	0.124	0.344
		(0.840–0.934)				(0.983–0.993)				(0.968–0.987)				(0.992–0.997)		
CALCA-VAL	0.900	0.978	0.135	0.374	2.017	0.998	0.086	0.239	2.618	0.997	0.127	0.354	2.493	0.991	0.232	0.644
		(0.965–0.986)				(0.997–0.999)				(0.996–0.998)				(0.987–0.995)		

Abbreviations: SD = standard deviation; ICC = intraclass correlation coefficient; SEM = standard error of measurement; MDC = minimal detectable change; CI = confidence interval; CALCA-ADDUC = calcaneus adduction; CALCA-ABDUC = calcaneus abduction; CALCA-PLANFLEX = calcaneus plantar flexion; CALCA-DORFLEX = calcaneus dorsal flexion; CALCA-VAR = calcaneus varus; CALCA-VAL = calcaneus valgus; BARE = barefoot; LWI4 = lateral wedge insoles, 4 mm; LWI7 = lateral wedge insoles, 7 mm; LWI10 = lateral wedge insoles, 10 mm.

**Table 5 jcm-11-04536-t005:** The data of the navicular bone sensor movement under the four conditions.

	BARE	LWI 4 mm	LWI7 mm	LWI 10 mm	*p*-Value BARE	*p*-Value BARE	*p*-Value BARE	*p*-Value LWI 4 mm	*p*-Value LWI 4 mm	*p*-Value LWI 7 mm
Variables	Mean (degrees)	Mean (degrees)	Mean (degrees)	Mean (degrees)	vs.	vs.	vs.	vs.	vs.	vs.
	± SD (95% CI)	± SD (95% CI)	± SD (95% CI)	± SD (95% CI)	LWI 4 mm	LWI 7 mm	LWI 10 mm	LWI 7 mm	LWI 10 mm	LWI 10 mm
NAVIC-ADDUC	0.51 ± 0.68	1.28 ± 2.18	1.50 ± 2.49	1.33 ± 2.27						
	(0.33–0.68)	(0.73–1.82)	(0.84–2.11)	(0.74–1.91)	0.2	0.08	0.22	0.3	0.76	2.67
NAVIC-ABDUC	0.66 ± 1.7	1.58 ± 2.63	1.76 ± 2.81	2.01 ± 2.95						
	(0.30–1.05)	(0.93–2.22)	(1.03–2.47)	(1.28–2.78)	0.06	0.06	<0.05 *	1.33	0.051	1.2
NAVIC-PLANFLEX	0.68 ± 1.34	3.02 ± 4.58	2.51 ± 3.55	2.48 ± 3.69						
	(0.38–0.98)	(2.04–4.36)	(1.60–3.42)	(1.54–3.42)	<0.001 **	<0.05 *	<0.001 **	2.4	1.26	2.7
NAVIC-DORFLEX	0.31 ± 0.46	1.04 ± 1.97	1.36 ± 2.43	0.56 ± 1.22						
	(0.19–0.42)	(0.53–1.53)	(0.78–1.94)	(0.26–0.88)	0.11	<0.05*	2.661	0.72	<0.05*	<0.05*
NAVIC-VAR	0.35 ± 0.95	1.23 ± 2.08	1.23 ± 1.60	0.98 ± 1.41						
	(0.11–0.60)	(0.87–1.7)	(0.84–1.58)	(0.62–1.50)	<0.001 **	<0.001 **	<0.001 **	0.66	2.31	0.15
NAVIC-VAL	0.27 ± 0.39	1.04 ± 1.98	0.96 ± 1.71	1.16 ± 1.73						
	(0.18–0.35)	(0.53–1.54)	(0.53–1.40)	(0.71–1.60)	<0.05 *	<0.05 *	<0.001 **	2.13	0.35	0.06

Abbreviations: SD = standard deviation; CI = confidence interval; NAVIC-ADDUC = navicular adduction; NAVIC-ABDUC = navicular abduction; NAVIC-PLANFLEX = navicular plantar flexion; NAVIC-DORFLEX = navicular dorsal flexion; NAVIC-VAR = navicular varus; NAVIC-VAL = navicular valgus; BARE = barefoot; LWI4 = lateral wedge insoles, 4 mm; LWI7 = lateral wedge insoles, 7 mm; LWI10 = lateral wedge insoles, 10 mm. *p*-Value = level of significance; *p* < 0.05 * (with a 95% confidence interval) was considered statistically significant, and *p* < 0.001 ** (with a 95% confidence interval) was considered strongly statistically significant.

**Table 6 jcm-11-04536-t006:** The data of the calcaneus bone sensor movement under the four conditions.

	BARE	LWI 4 mm	LWI7 mm	LWI 10 mm	*p*-Value BARE	*p*-Value BARE	*p*-Value BARE	*p*-Value LWI 4 mm	*p*-Value LWI 4 mm	*p*-Value LWI 7 mm
Variables	Mean (degrees)	Mean (degrees)	Mean (degrees)	Mean (degrees)	vs.	vs.	vs.	vs.	vs.	vs.
	± SD (95% CI)	± SD (95% CI)	± SD (95% CI)	± SD (95% CI)	LWI 4 mm	LWI 7 mm	LWI 10 mm	LWI 7 mm	LWI 10 mm	LWI 10 mm
CALCA-ADDUC	0.17 ± 0.68	1.76 ± 2.66	1.83 ± 2.42	1.91–2.81						
	(0.10–0.24)	(1.08–2.44)	(1.21–2.46)	(1.19–2.63)	<0.001 **	<0.001 **	<0.001 **	1.73	1.19	2.05
CALCA-ABDUC	0.55 ± 1.06	1.67 ± 2.79	1.60 ± 2.27	1.84 ± 3.00						
	(0.30–0.79)	(0.98–2.36)	(1.02–2.18)	(1.07–2.62)	<0.05 *	<0.05 *	<0.05 *	2.31	1.67	1.01
CALCA-PLANFLEX	0.35 ± 0.90	2.88 ± 4.31	3.07 ± 4.07	2.61 ± 3.56						
	(0.15–0.57)	(1.76–3.98)	(2.01–4.11)	(1.70–3.53)	<0.001 **	<0.001 **	<0.001 **	0.54	0.99	2.88
CALCA-DORFLEX	0.37 ± 0.79	1.60 ± 2.77	1.42 ± 2.34	1.36 ± 2.60						
	(0.18–0.56)	(0.89–2.31)	(0.81–2.02)	(0.69–2.02)	0.12	0.12	0.38	2.41	1.38	1.65
CALCA-VAR	0.20 ± 0.49	1.30 ± 1.70	1.21 ± 1.45	1.06 ± 1.69						
	(0.09–0.31)	(0.86–1.73)	(0.54–1.57)	(0.62–1.49)	<0.001 **	<0.001 **	<0.001 **	2.74	0.23	0.22
CALCA-VAL	0.40 ± 0.90	1.08 ± 2.01	1.74 ± 2.61	1.47 ± 2.50						
	(0.17–0.63)	(0.56–1.60)	(1.06–2.41)	(0.83–2.11)	0.19	<0.001 **	<0.001 **	<0.05 *	<0.05 *	2.55

Abbreviations: SD = standard deviation; CI = confidence interval; CALCA-ADDUC = calcaneus adduction; CALCA-ABDUC = calcaneus abduction; CALCA-PLANFLEX = calcaneus plantar flexion; CALCA-DORFLEX = calcaneus dorsal flexion; CALCA-VAR = calcaneus varus; CALCA-VAL = calcaneus valgus; BARE = barefoot; LWI4 = Lateral wedge insoles, 4 mm; LWI7 = lateral wedge insoles, 7 mm; LWI10 = lateral wedge insoles, 10 mm. *p*-Value = level of significance; *p* < 0.05 * (with a 95% confidence interval) was considered statistically significant, and *p* < 0.001 ** (with a 95% confidence interval) was considered strongly statistically significant.

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
