# Peer review of "Changes in the Kinematics of Midfoot and Rearfoot Joints with the Use of Lateral Wedge Insoles"

_jcm, 2022, doi:10.3390/jcm11154536_

Round 1

Reviewer 1 Report

Congratulations

Reviewer 2 Report

To the kind attention of the author

I think that the revised version of the article is suitable for publication

This manuscript is a resubmission of an earlier submission. The following is a list of the peer review reports and author responses from that submission.

Round 1

Reviewer 1 Report

Dear Authors,

the aim of the study is interesting, but I noticed too many flaws.

First, an extensive editing of English language is required. There are too many mistakes throughout the manuscript, especially for grammar (some verbs or asdjectives used have no sense at all such as "championed" at line 154) so it cannot be considered for publication at the moment.

Furthermore, table and figures are missing in the manuscript. Did you check for figures and tables before submitting the article?

Another mistake that I noticed at the end is that 52 references are reported in the manuscript, but only 51 references are present in the references list.

I would not reject the article since I think that, if revised appropiately, it could be of some interest for readers. 

Improve the use of English, put Tables and Figures, and make some part of the manuscript (such as the Material and Methods section) easier to understand.

Author Response

Responses to Reviewers' comments

Reviewer #1:

Thanks for your appraisals and comments in order to improve the quality of our manuscript. Deep and substantial modifications have been made according to your suggestions.

  1. In response to: make some part of the manuscript (such as the Material and Methods section) easier to understand.

Thank you for your appreciation. We are improving the manuscript following your indications. we have modified the text, lines 233-701

  1. In response to: Another mistake that I noticed at the end is that 52 references are reported in the manuscript, but only 51 references are present in the references list.

Thank you for your appreciation. We are improving the manuscript following your indications. we have modified the text.

  1. In response to: Furthermore, table and figures are missing in the manuscript. Did you check for figures and tables before submitting the article?

Thank you for your appreciation. We are improving the manuscript following your indications. we have added the figures and tables.

4.In response to:  Extensive editing of English language and style required

Thank you for your appreciation. We are improving the manuscript following your indications. We have sent it for review. we have modified the text.

We certify that the following article Kinematic effect on the navicular bone with the use of rearfoot valgus wedge. Álvaro Gómez Carrión, Maria de los Ángeles Atín Arratibe, Maria Rosario Morales Lozano, Carmen Martínez Rincón, Carlos Martínez Sebastián, Blanca De la Cruz-Torres, Álvaro Saura Sempere, Ruben Sanchez-Gomez * has undergone English language editing by MDPI. The text has been checked for correct use of grammar and common technical terms, and edited to a level suitable for reporting research in a scholarly journal. MDPI uses experienced, native English speaking editors. Full details of the editing service can be found at ► https://www.mdpi.com/authors/english.

Reviewer 2 Report

1. many typo errors

2. Few sentences need to corrected

3. Figure and table are missing from manuscript.

4. study limitations, strengths and weaknesses of study should be mentioned

Author Response

Responses to Reviewers' comments

Reviewer #2:

Thanks for your appraisals and comments in order to improve the quality of our manuscript. Deep and substantial modifications have been made according to your suggestions.

  1. In response to: typoerror

Thank you for your appreciation. We are improving the manuscript following your indications.

we have modified the text, lines 21,22,47

The minial changes are in the manuscript

  1. In response to: Correct the grammatical error

Thank you for your appreciations. We are improving the manuscript following your indications.

we have clarified and modified the text, lines 53-54

The Polhemus Fastrak Patriot, a three-dimensional electromagnetic device used in previous studies, will be used to measure the change of position of the navicular (22-25).

  1. In response to: too long sentence and difficult for reader. better to use 2-3 sentence.

Thank you for your appreciation. We are improving the manuscript following your indications.

we have modified the text, lines 21,22,47

Due to the interest in knowing the therapeutic effect in the investigation of RVLW in the foot, the objective of this study was to determine the response of the movements of the navicular in individuals with the use of three wedges rearfoot valgus.

  1. In response to: exclusion

Thank you for your appreciation. We are improving the manuscript following your indications. we have modified the text, lines 93

  1. In response to: determinate

Thank you for your appreciation. We are improving the manuscript following your indications. we have modified the text, lines 125

  1. In response to: name the software its volume etc

Thank you for your appreciation. We are improving the manuscript following your indications. we have modified the text, lines 134

FT3HostSWCD-2.1.0

7.In response to: follow uniform pattern for reporting mean and sd

Thank you for your appreciation. We are improving the manuscript following your indications. we have modified the text, lines 217-234

8.In response to: correct the citation

Thank you for your appreciation. We are improving the manuscript following your indications. we have modified the text.

9.In response to: study limitations, strengths and weaknesses of study should be mentioned

Thank you for your appreciation. We are improving the manuscript following your indications. we have modified the text.

Limitation

The study has some limitations, during the attachment of the sensors to the skin or when the wedges are changed, unwanted movements are generated. The subject takes time to adapt the wedges, it is possible that it would have generated instability during the data collection, even respecting the time to take the measurement. Sometimes a dispersion of data has been found in the sample and that is due to the high sensitivity of the instrument.

10.In response to:  Figure and table are missing from manuscript.

Thank you for your appreciation. We are improving the manuscript following your indications. we have added the figures and tables at the end of the word so as not to alter the enumeration

11.In response to:  Extensive editing of English language and style required

Thank you for your appreciation. We are improving the manuscript following your indications. Once the article has been reviewed, it will be sent to a translation service for the correction of the language, if it seems good to you.